# A Case-Crossover Study to Investigate the Effects of Atmospheric Particulate Matter Concentrations, Season, and Air Temperature on Accident and Emergency Presentations for Cardiovascular Events in Northern Italy

**DOI:** 10.3390/ijerph16234627

**Published:** 2019-11-21

**Authors:** Paolo Contiero, Roberto Boffi, Giovanna Tagliabue, Alessandra Scaburri, Andrea Tittarelli, Martina Bertoldi, Alessandro Borgini, Immacolata Favia, Ario Alberto Ruprecht, Alfonso Maiorino, Antonio Voza, Marta Ripoll Pons, Alessandro Cau, Cinzia DeMarco, Flavio Allegri, Claudio Tresoldi, Michele Ciccarelli

**Affiliations:** 1Environmental Epidemiology Unit, Fondazione IRCCS Istituto Nazionale dei Tumori, 20133 Milan, Italy; alessandra.scaburri@istitutotumori.mi.it (A.S.); martina.bertoldi@istitutotumori.mi.it (M.B.); alessandro.borgini@istitutotumori.mi.it (A.B.); immacolata.favia@istitutotumori.mi.it (I.F.); alessandro.cau@istitutotumori.mi.it (A.C.); 2Respiratory Disease Unit, Fondazione IRCCS Istituto Nazionale Tumori, 20133 Milan, Italy; roberto.boffi@istitutotumori.mi.it (R.B.); aaruprecht@gmail.com (A.A.R.); flavio.allegri@istitutotumori.mi.it (F.A.); 3Cancer Registry Unit, Fondazione IRCCS Istituto Nazionale dei Tumori, 20133 Milan, Italy; giovanna.tagliabue@istitutotumori.mi.it (G.T.); andrea.tittarelli@istitutotumori.mi.it (A.T.); 4International Society of Doctors for the Environment (ISDE), 52100 Arezzo, Italy; 5Pneumology Department, Humanitas Research Hospital, 20089 Rozzano, Italy; alfonso.maiorino@humanitas.it (A.M.); michele.ciccarelli@humanitas.it (M.C.); 6Emergency Department, Humanitas Research Hospital, 20089 Rozzano, Italy; antonio.voza@humanitas.it (A.V.); marta.ripoll_pons@humanitas.it (M.R.P.); 7Biomarkers Unit, Fondazione IRCCS Istituto Nazionale dei Tumori, 20133 Milan, Italy; cinzia.demarco@istitutotumori.mi.it; 8G. A. Maccacaro Unit of Medical Statistics, Biometry, and Epidemiology, Department of Clinical Sciences and Community Health, University of Milan, 20133 Milan, Italy; claudio.tresoldi@unimi.it

**Keywords:** particulate matter, atmospheric temperature, season, cardiovascular disease, climate change, accident and emergency, cancer, pollution, case-crossover study

## Abstract

Atmospheric particulate matter (PM) has multiple adverse effects on human health, high temperatures are also associated with adverse health outcomes, and the frequency of cardiovascular events (CVEs) varies with season. We investigated a hypothesized increase in PM-related accident and emergency (A&E) presentations for CVE with high temperature, warm season, days of high influenza incidence, and in people with a cancer diagnosis, using a time-stratified case-crossover study design. Outcomes were associations of A&E presentation for CVE with atmospheric PM ≤ 10 μm (PM_10_), season, and air temperature. PM_10_ levels in the municipality of residence (exposure variable) were estimated by modeling data from local monitoring stations. Conditional logistic regression models estimated odds ratios (OR) with 95% confidence intervals (CI) for presentations in relation to supposed influencers, adjusting for confounders. Study participants were all who presented at the A&E of a large hospital near Milan, Italy, for a CVE (ICD-9: 390–459) from 1st January 2014 to 31st December 2015. There were 1349 A&E presentations for CVE in 2014–2015 and 5390 control days. Risk of A&E presentation was significantly increased on hot days with OR 1.34 (95%CI 1.05–1.71) per 10 μg/m^3^ PM_10_ increment (as mean PM_10_ on day of presentation, and 1 and 2 days before (lags 0–2)), and (for lag 0) in autumn (OR 1.23, 95%CI 1.09–1.37) and winter (OR 1.18, 95%CI 1.01–1.38). Risks were also significantly increased when PM_10_ was on lag 1, in people with a cancer diagnosis in the spring and summer months (1.88, 95%CI 1.05–3.37), and on days (lags 0–2) of high influenza incidence (OR 2.34, 95%CI 1.01–5.43). PM_10_ levels exceeded the 50 μg/m^3^ “safe” threshold recommended by the WHO and Italian legislation for only 3.8% of days during the warm periods of 2014–2015. Greater risk of A&E presentation for CVE in periods of high PM_10_ and high temperature suggests that “safe” thresholds for PM_10_ should be temperature-dependent and that the adverse effects of PM_10_ will increase as temperatures increase due to climate change.

## 1. Introduction

Air pollution is a major public health problem. The Global Burden of Diseases, Injuries, and Risk Factors Study [1] identified fine (≤ 2.5 μm) atmospheric particulate matter (PM_2.5_) as responsible for 4.2 million deaths worldwide in 2015. Both PM ≤ 10 μm (PM_10_) and PM_2.5_ are known to have multiple adverse effects on human health [2,3] and are classified by the WHO and International Agency for Research on Cancer (IARC) as group 1 carcinogens (carcinogenic to humans) [4]. PM is emitted/formed by natural and anthropogenic processes. In industrial and urban areas, PM is mainly anthropogenic [5].

In 2015, there were an estimated 420 million cases of cardiovascular disease (CVD) worldwide, and around 18 million deaths [6]. In Italy, the age-standardized (World Population, 2012) death rate for both sexes together for cardiovascular diseases is 110.1 per 100,000 person-years, with 239,514 deaths in 2015 [7]. Extensive epidemiological and toxicological data demonstrate that particulate air pollution is associated with the development of CVD [8]. Short-term exposure can trigger acute events, while long-term exposure increases the risk of a cardiovascular event (CVE) and can reduce life expectancy [8]. A 2018 meta-analysis found positive associations between PM_10_ exposure and overall, cardiovascular and respiratory hospital admissions and mortality [9].

Associations between air temperature and CVEs and mortality have also been documented [10]. High air temperature has been associated with the worsened impact of PM_10_ on CVD outcomes including mortality [11,12,13]. A 2017 review [14], noted that CVD-related hospitalizations and mortality were linked to cold snaps in winter, heat waves in summer, and atmospheric pollution, but that the effect of season was complex.

CVEs can be investigated by accessing health databases such as hospital admissions/discharges. Emergency room or accident and emergency (A&E) databases are of particular interest however because they archive presentations for a broader range of CVEs than hospital discharges.

To our knowledge, no studies have investigated the effects of air temperature and season on the relation between PM_10_ and A&E presentations for CVEs. We hypothesized an increase in PM-related A&E presentations for CVE with high temperature, warm season, and on days of high influenza incidence. We also investigated a supposed interaction between PM_10_ and difference in temperature between the day of A&E presentation (or control day) and the day before, hypothesizing that a rapid change in temperature may have a more marked effect on the cardiovascular system than temperature alone. It is also likely that people who are frail—those with influenza and those with a previous cancer diagnosis—are susceptible to the combination of PM_10_ and temperature. We used a case-crossover approach to address these issues, investigating presentations for CVEs at the A&E unit of Humanitas Hospital, Rozzano, near Milan, in the plain of the river Po. We used PM_10_ measurements instead of PM_2.5_ because, at the start of the study, they were the only measurements available for the study period.

The plain of the Po is a densely-populated (355/km^2^) area of 46,000 square kilometers, enclosed by mountains to the north, west, and south. The mountains inhibit air circulation, and extended periods of air stasis contribute to some of the highest levels of air pollution in the world [15,16]. In Emilia Romagna—an administrative Region within the plain of the Po—the annual average concentrations of PM_10_ and PM_2.5_ exceeded the WHO limits of 20 and 10 μg/m^3^, respectively, and were estimated as responsible for 4.4 and 2.8 deaths per 100,000/year [17].

The Mediterranean region, within which the Po Valley is situated, was considered highly vulnerable to climate change in the 5th (2014) Assessment Report of the Intergovernmental Panel on Climate Change [18] and temperatures in the region are projected to increase in coming years [19].

## 2. Materials and Methods

### 2.1. Participants

A&E presentations from 1st January 2014 to 31st December 2015 were accessed from the Humanitas A&E database. Date of presentation, date of birth, sex, diagnosis, municipality of residence, and patient social security number were extracted. The study design was approved by the ethical committee of the Fondazione IRCCS Istituto Nazionale dei Tumori, Milan, on 28 November 2018. Only descriptive diagnoses were available—of which there could be several (those associated with the presentation and those for concomitant diseases, including cancer). From the descriptive diagnoses, presentations with a CVE were identified by project staff and transformed into ICD-9 codes (390–459). Presentations were linked, via social security number, with the Region of Lombardy Social Security List to ascertain vital status, and, where pertinent, date of death. Extracted data were managed in a study database with safeguards in place to prevent unauthorized access and respect patient privacy. The A&E presentations selected for the study had a CVE and a municipality of residence within 20 km of Humanitas hospital. In all, participants from 109 neighboring municipalities contributed to the study; the municipalities varied in size from a few hundred (population) to the population of Milan of about 1,340,000. The 20 km cut-off was adopted as it is known that residences distant from the hospital are more likely to be incorrect so that misclassification of exposure is more likely.

### 2.2. Exposure Assessment 

The Lombardy Environmental Protection Agency (ARPA) measures PM_10_ at a series of monitoring stations in the Region of Lombardy. Using a model developed in house, ARPA uses monitoring station data to produce estimates of mean daily PM_10_ for each municipality within the Region [20], taking account of emissions from local pollution sources, daily meteorological data, and orography. The PM_10_ exposure of each participant was considered to be that of their municipality of residence.

The air temperature and humidity to be assigned (to each participant) on days of presentation and non-presentation were obtained from the ARPA monitoring station nearest the municipality of residence [21]. Mean apparent temperature in degrees Celsius was assigned. Apparent temperature accounts for relative humidity effects and was estimated using the weather metrics procedure of the R statistical package [22].

### 2.3. Study Design and Statistical Analysis

To determine the risk of A&E presentation for a CVE in relation to mean daily PM_10_, a bidirectional time-stratified case-crossover design was used. Cases were persons presenting at the A&E for a CVE, controls were the same persons evaluated (for PM_10_, etc.) on days when they did not present. Control days were matched to case days by day of the week such that control days could be 7, 14, 21, or 28 days before or after the case day. Up to 4 control days per case were used.

PM_10_ exposure for each case and control was defined in 3 ways as follows: PM_10_ on the day of presentation (lag 0), PM_10_ on the day before (lag 1), PM_10_ two days before (lag 2).

Conditional logistic regression modeling was used to estimate odd ratios (ORs) for A&E presentation for a CVE, compared to non-presentation, in relation to PM_10_. A basic model assessed the effect of PM_10_ on CVE presentations considering various confounders. Three additional models were run with interactions for season, temperature and influenza incidence added successively, thus examining the effect on presentations of PM_10_ and season; PM_10_ season and temperature; and PM_10_ season temperature and influenza incidence.

The basic logistic model was:logit {Y = 1|X} = β_0_ + β_1_ PM_10_ + β_2_ flu + β_3_ holiday + β_4_ pop_dec + β_5_ app_temp + β_6_ ozone + β_7_ PM_10_ * sex + β_8_ PM_10_ * age
where Y is the binary response variable for A&E presentation that takes values of 1 or 0, the Xs are covariates and the βs regression coefficients. PM_10_ was considered as a linear variable, and ORs were estimated for 10 μg/m^3^ increments of PM_10_. Dummy variables for influenza incidence (flu), day of holiday (holiday), and population decrease due to summer holidays (pop_dec) were added. Days of influenza incidence were identified by consulting Influnet, the Italian national surveillance system [23,24] and categorized into four levels, from no influenza days to high influenza days.

A term for apparent temperature (app_temp) was also included, categorized as: high (>80th percentile), medium (20th–80th percentiles), and low (<20th percentile). A linear term for ozone levels (obtained from ARPA monitoring stations) and interaction terms of PM_10_, with sex and age were also added because of their supposed roles as confounders or modifiers.

Separate basic models were run for PM_10_ at lags 0, 1, 2 and 0–2 (the latter as average PM_10_ at lags 0, 1 and 2). The linearity of the relationship between A&E presentation and PM_10_ was assessed by adding a restricted cubic spline term to model PM_10_ and evaluating the difference between models with and without the restricted cubic spline, using the likelihood ratio test [25].

A product term, PM_10_*season, was added to the basic model to test whether season modified the association between PM_10_ and CVE presentation and the likelihood ratio test used to compare models with and without the PM_10_*season interaction term. Seasons were defined as 21 March–20 June (spring), 21 June–22 September (summer), 23 September–20 December (autumn), and 21 December–20 March (winter). Separate models were run at lags 0, 1, 2 and 0–2.

A product term, PM_10_*apparent_temperature, was added to the basic plus season model to test whether apparent temperature modified the association between PM_10_ and CVE presentation, the likelihood ratio test was used to compare models with and without the PM_10_ and apparent temperature interaction term. Separate models were run at lags 0, 1, 2, and 0–2.

We also hypothesized the interaction between PM_10_ and the difference in temperature between the day of A&E presentation (or control day) and the day before. This was tested by computing the difference between the temperature of the day of A&E presentation (or control day) and the day before and inserting into the model a term representing this difference in temperature (diff-temperature) and a product term (PM_10_*diff-temperature) representing the interaction of diff-temperature with PM_10_. We modeled diff-temperature using the lsp function of R-language that fits a linear spline, with a knot placed at −1 °C, i.e., a bilinear function that decreases linearly to −1 °C and increases linearly from −1 °C.

Finally, the product term PM_10_*influenza was added to the basic plus season plus apparent temperature model, and the likelihood ratio test used to test for an interaction between PM_10_ and influenza in relation to the A&E presentation.

Separate models (basic and seasonal models) that included only patients (A&E presentations) with a diagnosis of cancer were run. Because of the small number of cancer patients, only two seasons were considered: cold season (autumn plus winter) and warm season (spring plus summer).

The analyses were performed using the R statistical package [22], version 3.6.1. Differences were considered significant for *p* < 0.05.

## 3. Results

A total of 1349 A&E presentations for CVE were identified in the 2014–2015 study period, with 5390 control days were selected for comparison (thus most presentation dates for most participants had four control dates). Table 1 shows the distributions of presentations by age and sex, together with the distributions of PM_10_ and apparent temperature, all by season. A total of 168 presenters had been previously diagnosed with cancer (as determined from the A&E database).

Table 2, Table 3, Table 4 and Table 5 show odds ratios produced by the models for lag 0, lag 1, lag 2, and lags 0–2, respectively. The second column of each of these tables shows results for the basic models: PM_10_ was never significant in any model or any lag. The third column of these tables shows results when the PM_10_*season interaction was added to the basic model. With spring as reference, ORs in autumn for A&E presentations for CVEs were 1.21 (95%CI 1.09–1.35) per 10 μg/m^3^ increment in PM_10_ for lag 0, 1.14 (95%CI 1.02–1.27) for lag 1, 1.05 (95%CI 0.94–1.17) for lag 2, and 1.17 (95%CI 1.04–1.33) for lags 0–2. ORs for A&E presentations in winter (reference spring) were 1.13 (95%CI 1.02–1.26) for lag 0, 1.15 (95%CI 1.04–1.28) for lag 1, 1.06 (95%CI 0.96–1.18) for lag 2, and 1.16 (95%CI 1.03–1.31) for lags 0–2. ORs for summer were never significant for any lag.

The fourth column of Table 2, Table 3, Table 4 and Table 5 shows results after adding the interaction between PM_10_ and apparent temperature to the model of the third column. In this model ORs per season changed slightly compared to the previous model, but significance never changed. When it was hot (reference moderate temperature), ORs of presentation were 1.18 (95%CI 0.97–1.45) per 10 μg/m^3^ PM_10_ increment for lag 0, 1.28 (95%CI 1.04–1.57) for lag 1, 1.23 (95%CI 1.00–1.51) for lag 2, and 1.34 (95%CI 1.04–1.71) for lags 0–2. ORs were never significant when it was cold. Introduction of the terms diff-temperature and PM_10_* diff-temperature never had a significant effect on any model for any lag.

The fifth column of Table 2 shows results after adding an interaction term for PM_10_ and influenza to the model of the fourth column. In this model, ORs for season changed slightly compared to the previous model but none of the significant ORs became non-significant and none of the non-significant ORs became significant. On days when influenza notifications were high, (reference days with no influenza) ORs per 10 μg/m^3^ PM_10_ increment were 1.15 (95%CI 0.73–1.81) for lag 0, 1.34 (95%CI 0.80–2.25) for lag 1, 1.35 (95%CI 0.84–2.17) for lag 2, and 2.34 (95%CI 1.01–5.42) for lags 0–2. After inserting the interaction term between PM_10_ and influenza incidence, the interaction between PM_10_ and winter became non-significant for lags 0–2.

The use of restricted cubic spline PM_10_ modeling and the likelihood ratio test provided no reason to reject the null hypothesis of a linear association between A&E presentation and PM_10_ levels. Similar testing provided no reason to reject the null hypothesis of a linear association between A&E presentation and PM_10_ levels on days of high influenza incidence.

For presenters with a previous cancer diagnosis, the basic model provided ORs of 1.03 (95%CI 0.81–1.32) for lag 0, 1.08 (95%CI 0.81–1.43) for lag 1, 1.21 (95%CI 0.90–1.63) for lag 2, and 1.13 (95%CI 0.83–1.54) for lags 0–2. For presenters with a previous cancer diagnosis, ORs for warm season presentation were 1.64 (95%CI 0.89–3.01) for lag 0, 1.88 (95%CI 1.05–3.37) for lag 1, 1.32 (95%CI 0.80–2.19) for lag 2, and 1.13 (95%CI 0.83–1.54) for lags 0–2. For presenters with a previous cancer diagnosis, ORs for cold season presentation were 0.92 (95%CI 0.69–1.25) for lag 0, 0.84 (95%CI 0.57–1.23) for lag 1, 1.15 (95%CI 0.80–1.67) for lag 2, and 0.93 (95%CI 0.62–1.40) for lags 0–2 (data not presented in tables). In none of the models was there a significant interaction between PM_10_ and sex or age.

ARPA modeling estimated that PM_10_ levels exceeded the 50 μg/m^3^ “safe” threshold recommended by the WHO [26] and Italian legislation for only 3.8% of days (in some municipalities) during the warm periods of 2014–2015.

## 4. Discussion

We have found that increasing atmospheric PM_10_ (continuous variable) was associated with greater risk of A&E presentation for a CVE during autumn and winter than spring, and also when temperatures were high (reference moderate temperature) irrespective of the season. PM_10_ was not associated with A&E presentation for a CVE in summer or on cold days. Thus our analyses distinguished the effect of season from the effect of temperature. We also investigated whether a rapid change in temperature from the day before the day of presentation interacted with PM_10_, but found no such interaction. We also found that the risk of presentation increased significantly with increasing PM_10_ during days of high influenza incidence, which occurred exclusively in the cold winter months. Furthermore, for persons with a previous cancer diagnosis—who are to be considered frail according to standard definitions [27]—the risk of A&E presentation for CVE increased with increasing PM_10_ during spring plus summer, but not during autumn plus winter.

To our knowledge, no previous study has reported the risk of A&E presentation for CVEs in relation to interactions between PM_10_ and season and interactions between PM_10_ and temperature at the same time. However, this important novelty renders comparisons with the findings of other studies difficult. In fact, few previous studies have investigated the influence of temperature on PM–related A&E presentations for CVEs. A 2017 study conducted in Beijing [12] found that CVEs associated with 10 mg/m^3^ increments of PM_10_ increased by 0.14% when air temperature was ≥28 °C. A study conducted in Brisbane [11] found that high PM_10_ was associated with more adverse health effects, including cardiovascular emergency visits, on warm days than cold days, for lags up to 2 days. Our findings are fully consistent with these findings. They are also consistent with the findings of a 2018 study on a European urban area [28] that investigated total natural and cardiovascular mortality in relation to air temperature and particle number concentration, PM_2.5_, PM_10_, and ozone. This study found that on high air temperature days (>75th percentile), a 10 μg/m^3^ increase in PM_10_ was associated with a 1.61% increase in cardiovascular mortality, which was significantly higher than on low temperature (<25th percentile) days. Furthermore, on high air pollution days (>50th percentile), both hot- and cold-related mortality risks increased.

The modulation of the effect of PM_10_ on CVE admissions with season and temperature could be due, at least in part, to seasonal variation in the chemical composition of PM_10_. Thus, a study performed in the Po Valley (site of our study) found that PM_10_ levels of ammonium, elemental carbon and organic carbon were higher during winter, while levels of aluminum, silicon, magnesium, and calcium were higher in summer [29]. Furthermore, levels of chromium and mercury in PM_10_ at a roadside site in Beijing were found to be higher in summer than winter [30].

A 2013 study [31] exposed human macrophage-like cells to summer and winter PM_10_ from the Milan atmosphere and measured the release of the proinflammatory interleukin 1β (IL1β). The study found that IL1β release increased in a dose-dependent manner on exposure to summer PM_10_ but not to winter PM_10_. The authors concluded that Milan summer PM_10_ contains substances that promote the activation of membrane Toll-like receptors and the NLPR3 inflammasome, to stimulate IL1β release. We found no risk increase in summer but did find a risk increase associated with high temperature, which could, we suggest, be related to chemical components present in summer PM_10_, that might be particularly high at times of high temperature.

The higher risk of CVE presentation with high PM_10_ in winter and autumn also suggests that behavioral changes determined by seasonal factors might be playing a role. The review of Steward et al. [12] noted that seasonal behavioral changes in diet and physical activity influenced CVE risk which could be mediated by an effect on PM-induced inflammation.

High temperatures exacerbate the toxic effects of many environmental toxins by various mechanisms [13]. PM_10_, and its toxic components, enter the body via the skin and respiratory surfaces. When air temperature is high peripheral vasodilatation increases, resulting in higher skin temperature. Respiration is also moderately increased to increase heat dissipation by evaporation. These two temperature responses enhance the entry of PM into the body.

Toxin inhalation may result in pulmonary inflammation and oxidative stress, leading to the release of proinflammatory cytokines giving rise to systemic inflammation. People typically spend more time outdoors and open their windows more during the summer months, thereby increasing their exposure to PM and other atmospheric pollutants [32].

It is noteworthy that in our study, A&E presentations for CVEs increased significantly with increasing PM_10_ during days of high influenza incidence. Influenza has been implicated as a contributory factor to acute myocardial infarction [33,34]. The increased levels of inflammatory cytokines, in particular, IL18 [35], that occur as a response to viral influenza may increase the risk myocardial infarction. Increased inflammatory cytokine levels due to high PM_10_ may further increase the risk of infarction.

People with a previous cancer diagnosis are frail [27] and known to be at increased risk of hospital admission for several causes not directly related to cancer [32]. Frailty is a state of heightened vulnerability to stressors [27]. In our case stressors would include cancer, treatment-induced cardiotoxicity, PM and air temperature [36]. In fact, CVD is the second leading cause of morbidity and mortality in cancer survivors mainly because of treatment-induced cardiotoxicity [37]. PM_10_ also influences cardiovascular mortality [7,28] and cancer prognosis [38]. We found that ORs for A&E presentations for CVE were higher for cancer patients than other categories, highlighting cancer patient susceptibility to high PM_10_.

We assessed risks of presentation in relation to lag between high PM_10_ and day of presentation. In winter and autumn, the effect of high PM_10_ on the risk of presentation was numerically greatest on the day of exposure (lag 0) to declined thereafter. By contrast, the effects of hot day and high flu incidence on PM_10_-related presentation seemed to be cumulative since ORs were greatest for lag 0–2. This variation in risk of PM_10_-related CVE presentation with lag suggests that various mechanisms are involved.

Our study has limitations. First, the study population is fairly small. Second, each person’s PM_10_ exposure was considered to be the mean daily exposure estimated for the entire municipality of residence, since only municipality-level PM_10_ estimates were available. However, misclassifications of exposure due to municipality-level estimation are likely to weaken associations overall rather than introduce bias. A further study limitation is that we only used PM_10_ and not other PM fractions such as PM_2.5_.

The case-crossover design is a study strength since confounding due to quasi-periodic fluctuations in pollution levels over a week were controlled for by day-of-the-week matching, and time-invariant confounders like age, sex, and smoking were controlled automatically. The persons presenting at the A&E with CVEs were unselected so our sample is likely to be representative of the general population. Furthermore the PM_10_ measurement method and the diagnosis assignments (ICD 9 codes) were standard. Thus we expect the generalizability of our findings to be good.

## 5. Conclusions

This study provides evidence that increasing temperature exacerbates at least one of the documented adverse effects of atmospheric PM_10_ on human health. However European and Italian legislation indicating “safe” thresholds for atmospheric PM_10_ levels does not take account of this exacerbating effect. Our data suggest—at the very least—that these thresholds should be lower during periods of high air temperature. Measures to ensure that thresholds are not exceeded are also required. As global temperatures increase [15,16] the adverse health effects of atmospheric PM_10_ will probably worsen, irrespective of whether atmospheric PM_10_ levels are reduced. It would also be advisable to institute systems to warn residents of high air pollution days so that vulnerable people, e.g., immunocompromised and elderly people could stay at home. Finally, we suggest a study to sample PM and analyze their components to further investigate the link between PM and A&E presentations for CVE.

## Figures and Tables

**Table 1 ijerph-16-04627-t001:** Patient characteristics, atmospheric particulate matter ≤10 μm (PM_10_), apparent atmospheric temperature and incidence of influenza by season.

	Season			
	Spring	Summer	Autumn	Winter
**Distribution of A&E presentations (%)**	28.8	22.7	19.8	28.7
**Age distribution (years)**				
Median (IQR)	78 (67–84)	76 (63–83)	78 (68–85)	80 (71–86)
**Sex (%)**				
Male	56.6	58.4	58.5	54.3
Female	44.4	41.6	41.5	45.7
**PM_10_** (μg/m^3^)				
Median (IQR)	24 (17–34)	23 (17–28)	38 (29–54)	47 (31–59)
**Apparent temperature** (°C)				
Median (IQR)	17 (14–20)	22 (20–25)	13 (10–16)	6 (4–8)
**Influenza incidence (as % of days)**				
Absent	95	100	96	13
Low	5	0	4	39
Medium	0	0	0	43
High	0	0	0	5

**Table 2 ijerph-16-04627-t002:** Odds ratios (OR) with 95% confidence intervals (CI) for accident and emergency (A&E) presentations for cardiovascular events, for lag 0, in relation to 10 μg/m^3^ PM_10_ increments, according to season, temperature, and influenza incidence.

Model Variables	OR (95%CI)			
	Basic Model (PM_10_)	PM_10_ + Season Interaction	PM_10_ + Season + Temperature Interaction	PM_10_ + Season + Temperature + Influenza Interaction
**PM_10_**	0.96 (0.86–1.07)	0.83 (0.72–0.96)	0.91 (0.66–1.25)	0.92 (0.67–1.28)
**PM_10_ and season interaction:**				
PM_10_ and spring	N/A	1 (ref)	1 (ref)	1 (ref)
PM_10_ and summer	N/A	1.11 (0.93–1.33)	1.06 (0.87–1.28)	1.05 (0.86–1.27)
PM_10_ and autumn	N/A	1.21 (1.09–1.35)	1.23 (1.10–1.38)	1.22 (1.09–1.37)
PM_10_ and winter	N/A	1.13 (1.02–1.26)	1.15 (1.02–1.29)	1.20 (1.02–1.40)
**PM_10_ and temperature interaction:**				
PM_10_ and mod. temp.	N/A	N/A	1 (ref)	1 (ref)
PM_10_ and cold temp.	N/A	N/A	1.01 (0.92–1.10)	0.99 (0.90–1.10)
PM_10_ and hot temp.	N/A	N/A	1.18 (0.97–1.45)	1.19 (0.97–1.46)
**PM_10_ and flu incidence interaction:**				
PM_10_ and no flu	N/A	N/A	N/A	1 (ref)
PM_10_ and low flu	N/A	N/A	N/A	0.91 (0.79–1.06)
PM_10_ and medium flu	N/A	N/A	N/A	0.97 (0.82–1.14)
PM_10_ and high flu	N/A	N/A	N/A	1.15 (0.73–1.81)

N/A = not applicable.

**Table 3 ijerph-16-04627-t003:** Odds ratios (OR) with 95% confidence intervals (CI) for A&E presentations for cardiovascular events, for lag 1, in relation to 10 μg/m^3^ PM_10_ increments, according to season, temperature and influenza incidence.

Model Variables	OR (95%CI)			
	Basic Model (PM_10_)	PM_10_ + Season Interaction	PM_10_ + Season + Temperature Interaction	PM_10_ + Season + Temperature + Influenza Interaction
**PM_10_**	1.03 (0.92–1.16)	0.92 (0.79–1.06)	0.99 (0.73–1.34)	0.99 (0.73–1.34)
**PM_10_ and season interaction:**				
PM_10_ and spring	N/A	1 (ref)	1 (ref)	1 (ref)
PM_10_ and summer	N/A	1.18 (0.99–1.41)	1.07 (0.88–1.31)	1.07 (0.88–1.31)
PM_10_ and autumn	N/A	1.14 (1.02–1.27)	1.16 (1.04–1.30)	1.16 (1.04–1.30)
PM_10_ and winter	N/A	1.15 (1.04–1.28)	1.17 (1.04–1.32)	1.16 (1.00–1.36)
**PM_10_ and temperature interaction:**				
PM_10_ and mod. temp.	N/A	N/A	1 (ref)	1 (ref)
PM_10_ and cold temp.	N/A	N/A	1.01 (0.93–1.10)	0.99 (0.90–1.09)
PM_10_ and hot temp.	N/A	N/A	1.28 (1.04–1.57)	1.28 (1.04–1.57)
**PM_10_ and flu incidence interaction:**				
PM10 and no flu	N/A	N/A	N/A	1 (ref)
PM10 and low flu	N/A	N/A	N/A	0.99 (0.86–1.14)
PM10 and medium flu	N/A	N/A	N/A	1.03 (0.88–1.20)
PM10 and high flu	N/A	N/A	N/A	1.34 (0.80–2.25)

N/A = not applicable.

**Table 4 ijerph-16-04627-t004:** Odds ratios (OR) with 95% confidence intervals (CI) for A&E presentations for cardiovascular events, for lag 2, in relation to 10 μg/m^3^ PM_10_ increments, according to season, temperature and influenza incidence.

Model Variables	OR (95%CI)			
	Basic Model (PM_10_)	PM_10_ + Season Interaction	PM_10_ + Season + Temperature Interaction	PM_10_ + Season + Temperature + Influenza Interaction
**PM_10_**	1.09 (0.97–1.23)	1.05 (0.91–1.21)	1.14 (0.87–1.51)	1.12 (0.85-1.47)
**PM_10_ and season interaction:**				
PM_10_ and spring	N/A	1 (ref)	1 (ref)	1 (ref)
PM10 and summer	N/A	1.02 (0.86–1.21)	0.94 (0.77–1.13)	0.96 (0.79–1.21)
PM10 and autumn	N/A	1.05 (0.94–1.17)	1.07 (0.95–1.19)	1.08 (0.97–1.21)
PM10 and winter	N/A	1.06 (0.96–1.18)	1.09 (0.97–1.23)	1.02 (0.88–1.19)
**PM_10_ and temperature interaction:**				
PM10 and mod. temp.	N/A	N/A	1 (ref)	1 (ref)
PM10 and cold temp.	N/A	N/A	0.98 (0.90–1.07)	0.97 (0.88–1.07)
PM10 and hot temp.	N/A	N/A	1.23 (1.00–1.51)	1.25 (1.01–1.55)
**PM_10_ and flu incidence interaction:**				
PM10 and no flu	N/A	N/A	N/A	1 (ref)
PM10 and low flu	N/A	N/A	N/A	1.11 (0.96–1.27)
PM10 and medium flu	N/A	N/A	N/A	1.09 (0.93–1.27)
PM10 and high flu	N/A	N/A	N/A	1.35 (0.84–2.17)

N/A = not applicable.

**Table 5 ijerph-16-04627-t005:** Odds ratios (OR) with 95% confidence intervals (CI) for A&E presentations for cardiovascular events, for lag 0–2, in relation to 10 μg/m^3^ PM_10_ increments, according to season, temperature and influenza incidence.

Model Variables	OR (95%CI)			
	Basic Model (PM_10_)	PM_10_ + Season Interaction	PM_10_ + Season + Temperature Interaction	PM_10_ + Season + Temperature + Influenza Interaction
**PM_10_**	1.03 (0.90.1.18)	0.90 (0.76–1.06)	0.99 (0.73–1.03)	0.99 (0.70–1.40)
**PM_10_ and season interaction:**				
PM_10_ and spring	N/A	1 (ref)	1 (ref)	1 (ref)
PM10 and summer	N/A	1.14 (0.93–1.41)	1.03 (0.82–1.29)	1.03 (0.82–1.29)
PM10 and autumn	N/A	1.17 (1.04–1.33)	1.20 (1.06–1.37)	1.21 (1.06–1.37)
PM10 and winter	N/A	1.16 (1.03–1.31)	1.19 (1.03–1.36)	1.17 (0.98–1.39)
**PM_10_ and temperature interaction:**				
PM10 and mod. temp.	N/A	N/A	1 (ref)	1 (ref)
PM10 and cold temp.	N/A	N/A	1.00 (0.91–1.11)	0.98 (0.88–1.10)
PM10 and hot temp.	N/A	N/A	1.34 (1.04–1.71)	1.34 (1.04–1.71)
**PM_10_ and flu incidence interaction:**				
PM10 and no flu	N/A	N/A	N/A	1 (ref)
PM10 and low flu	N/A	N/A	N/A	1.01 (0.85–1.19)
PM10 and medium flu	N/A	N/A	N/A	1.04 (0.87–1.26)
PM10 and high flu	N/A	N/A	N/A	2.34 (1.01–5.42)

N/A = not applicable.

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
