# Peer review of "A Case-Crossover Study to Investigate the Effects of Atmospheric Particulate Matter Concentrations, Season, and Air Temperature on Accident and Emergency Presentations for Cardiovascular Events in Northern Italy"

_ijerph, 2019, doi:10.3390/ijerph16234627_

Round 1
Reviewer 1 Report
Summary:
The manuscript addresses the association between atmospheric particulate matter (PM10) and cardiovascular events (CVE) near Milan Italy. The authors saw an increased risk of CVE per 10 ug/m3 increase in PM10. The authors observed significant increases in CVE events on autumn days in winter and saw some increased risk among those with cancer diagnoses and when there was a high incidence of influenza. The authors conclude that high temperatures contribute to CVE and that adverse effects of PM10 could worsen with climate change.
Suggestion: Accept with revisions
· Title could indicate location of study (i.e. Italy, or even more specific)
· Introduction
o Could you mention the direct impact of PM10 on adverse health? How many deaths are attributable to PM10 rather than PM2.5?
o A discussion of CVE and incidence in Italy would help readers understand the magnitude of the problem in Italy
o Additional information on typical ranges of air pollution would also be helpful. Seasonal averages would be very interesting if they are available. Without these two pieces of information on CVEs and air pollution, I do not know the magnitude of the issue in Italy
o Is there any difference in “accident” and “emergency” CVEs? Are there differences in biological mechanisms?
o Line 79, you say the area is densely population. Again, can you provide numbers here? How many people live here?
o I think the introduction is significantly lacking information and connection between the study exploring cancer and flu incidence with CVEs. Why is this important? What biological mechanisms underpin the connection between PM and CVE for these two conditions? Why did you explore this?
· Methods:
o When staff identified diagnoses and presentations of CVE, was this a group discussion or a person was the primary coder and this was checked by additional staff (i.e. reliability of diagnosis)?
o I do not know anything about Italy’s municipalities. How big are these areas? How many municipalities are represented in the study?
o Line 158: You can site the R program. Sometimes authors choose to site the specific R package used as well.
· Tables:
o Table 1 is very difficult to read. Why did you choose to separate into quartiles on the side? It might look cleaner if you just present the median and IQR for each season. That would work for continuous variables of age and PM.
o Table 2 is also very long. Maybe split into different lag versions so the table headings are easy to see on every page?
o In the description after the tables, I don’t think it’s necessary to refer to it as “in the third column…in the fourth column”.
· Discussion:
o The discussion and conclusions of temperature vs. season is not complete. Associations were observed in autumn and winter, and on hot days (but not summer). I think there is another story to tell here. Do you think that this is a result of “abnormalities” in temperature? We expect temperature to be higher in summer, but your results so no indication that summer hot days played a role. In autumn and winter when temperatures are relatively lower, does the impact of a very hot day (a more dramatic change in temperature) play a role? I think you are missing the explanation as to why we didn’t see any associations for summer, even though those are what we typically assume are “hot” days.
§ Is there any geographical reason why high temperatures would have an influence in this region?
o Lines 243-245: I am not sure that the study in Beijing is all that relevant to Milan unless the source of PM10 could be found in Milan. Do you think it is?
o Lines 246-250. The study you reference saw dose-dependent proinflammatory release in summer but not winter. This is the opposite of your results; why did you reference it here? If you mean it’s due to high temperature, that’s where the significance and reasoning between season and temperature is blurred and confusing.
o Lines 266-270. This seems to be where the heart of your discussion should be. I do not think these four lines are a sufficient analysis of your results. Many factors, behavioral, seasonal sources, or the impact of extreme weather that is “out of season” would be appropriate here. Authors should spend much more timing thinking about why these results were observed, or produce follow up questions.
o The discussion around cancer and flu is fine, but this needs to be discussed in the introduction.
o Lines 285: The importance of the lag is mentioned here. What other mechanisms do you think are important here? Again, the biological mechanisms of temperature on CVE should be explored in more depth.
· Conclusions
o The conclusion that the “safe” levels of air pollution are set too high is very important. I think that in the results that only 3.8% of days were above the threshold should be mentioned in the abstract.
o Is there an alert system in Italy that warns citizens of high air pollution days? (Or moderate air pollution?) Do they take into account vulnerable populations (immunocompromised, elderly, etc.) If not, this could also be a suggestion for public health officials.
Author Response
“Title could indicate location of study (i.e. Italy, or even more specific)”:
The title now refers to the study location; northern Italy
Introduction
- Could you mention the direct impact of PM10 on adverse health? How many deaths are attributable to PM10 rather than PM2.5?
We now state how many deaths are estimated attributed to PM for an administrative Region (Emilia Romagna) within the plain of the Po (line 107 et seq.)
- A discussion of CVE and incidence in Italy would help readers understand the magnitude of the problem in Italy
Only the CVE mortality rate is available: we now cite this rate in the Introduction, (line 71 et seq.)
- Additional information on typical ranges of air pollution would also be helpful. Seasonal averages would be very interesting if they are available. Without these two pieces of information on CVEs and air pollution, I do not know the magnitude of the issue in Italy”
We state in the Introduction that the study was carried out at a hospital in the plain of the river Po, northern Italy, characterized by some of the highest levels of air pollution in the world [ref 164]. Table 1 shows median (IQR) PM10 levels (μg/m3) by season.
- Is there any difference in “accident” and “emergency” CVEs? Are there differences in biological mechanisms
“Accident and Emergency, (A&E)” is the term used in English-speaking Europe. It is entirely equivalent to “Emergency Room” or “Emergency Unit” – terms in common use in North America.
- Line 79, you say the area is densely population. Again, can you provide numbers here? How many people live here?”
Population density in the plain of Po is now stated (355 persons/km2) Line 104.
- I think the introduction is significantly lacking information and connection between the study exploring cancer and flu incidence with CVEs. Why is this important? What biological mechanisms underpin the connection between PM and CVE for these two conditions? Why did you explore this”
The underlying idea here is that people with influenza – like those with a cancer diagnosis – are “frail” and thus more susceptible to a CVE in conditions of to the high of PM and high temperature. Now added to lines 91-99 (Introduction). We have already elaborated on the concept of frailty in the Discussion
Methods
- “When staff identified diagnoses and presentations of CVE, was this a group discussion or a person was the primary coder and this was checked by additional staff (i.e. reliability of diagnosis)?”
The diagnoses were performed by A&E physicians. The diagnostic part of the text was written by an A&E physician and reviewed by colleagues. The coding of the diagnoses was performed by the epidemiological team with two persons coding and a physician checking coding quality. For the information of the reviewer: we have not changed the text.
- “I do not know anything about Italy’s municipalities. How big are these areas? How many municipalities are represented in the study?”
In the “Participants” paragraph a sentence was added to explain municipality features, rows 133-135
- “Line 158: You can site the R program. Sometimes authors choose to site the specific R package used as well.”
We now specify that R version 3.6.1 was used.
Tables
- “Table 1 is very difficult to read. Why did you choose to separate into quartiles on the side? It might look cleaner if you just present the median and IQR for each season. That would work for continuous variables of age and PM.”
Table 1 was rebuild according to the suggestion of the reviewer
- “Table 2 is also very long. Maybe split into different lag versions so the t able headings are easy to see on every page?”
To meet these criticisms we have split the information into 4 separate tables (new tables 2 to 5). Each table presents data for a different lag (lag 0, lag 1, lag 2, lags 0-2).
Discussion
The discussion and conclusions of temperature vs. season is not complete. Associations were observed in autumn and winter, and on hot days (but not summer). I think there is another story to tell here. Do you think that this is a result of “abnormalities” in temperature? We expect temperature to be higher in summer, but your results so no indication that summer hot days played a role. In autumn and winter when temperatures are relatively lower, does the impact of a very hot day (a more dramatic change in temperature) play a role? I think you are missing the explanation as to why we didn’t see any associations for summer, even though those are what we typically assume are “hot” days”
We have performed new analyses introducing terms into the models to take account of abrupt temperature changes and a supposed interaction with PM10. These terms had no significant effects (Discussion line 312 et seq; Results line 275 et seq; Methods line 197 et seq). We feel that the new analysis renders our paper more informative. See Discussion Rows 196-206
Is there any geographical reason why high temperatures would have an influence in this region?
The orography of the plain of Po favors the development of temperature inversions leading to extended periods of atmospheric stability that impede the dispersion of PM. We have found that high exposure to PM interacts with high temperature and propose reasons for this in the discussion. We do not propose any direct geographic effect of high temperature. We have not changed the text.
Lines 243-245: I am not sure that the study in Beijing is all that relevant to Milan unless the source of PM10 could be found in Milan. Do you think it is?”
We believe that the findings of the Beijing study are pertinent in a general way and deserve to be cited, particularly since Beijing is now one of the most polluted cities on the planet.
Lines 246-250. The study you reference saw dose-dependent proinflammatory release in summer but not winter. This is the opposite of your results; why did you reference it here? If you mean it’s due to high temperature, that’s where the significance and reasoning between season and temperature is blurred and confusing.”
We have re-written this part of the Discussion to better explicate the link between this study and our study. In particular we now state: “We found no risk increase in summer but did find a risk increase associated with high temperature, which could, we suggest, be related to chemical components present in summer PM10, that might be particularly high at times of high temperature.
Lines 266-270. This seems to be where the heart of your discussion should be. I do not think these four lines are a sufficient analysis of your results. Many factors, behavioral, seasonal sources, or the impact of extreme weather that is “out of season” would be appropriate here. Authors should spend much more timing thinking about why these results were observed, or produce follow up questions.
We have rewritten the Discussion (lines 340 et seq.) to accommodate these criticisms.
- The discussion around cancer and flu is fine, but this needs to be discussed in the introduction.”
Cancer and flu are now mentioned briefly in the Introduction.
- Lines 285: The importance of the lag is mentioned here. What other mechanisms do you think are important here? Again, the biological mechanisms of temperature on CVE should be explored in more depth.”
We have rewritten the Discussion (lines 340 et seq.) to accommodate these criticisms.
· Conclusions
The conclusion that the “safe” levels of air pollution are set too high is very important. I think that in the results that only 3.8% of days were above the threshold should be mentioned in the abstract.
We have added this point to the abstract,
Is there an alert system in Italy that warns citizens of high air pollution days? (Or moderate air pollution?) Do they take into account vulnerable populations (immunocompromised, elderly, etc.) If not, this could also be a suggestion for public health officials.”
We now suggest in the Conclusions that the authorities should warn residents about high pollution days.
Reviewer 2 Report
The authors describe their work on a case-crossover study to investigate the effects of atmospheric particulate matter (PM) concentrations, season and air temperature on accident and emergency presentations for CVEs. It was found that greater risk A&E presentation for CVE occurred in periods of high PM and high temperature. It was concluded that safe thresholds for PM should be temperature dependent and that the adverse effects of PM will increase as temperature increases due to climate change. This is an interesting study. Appropriate methodology has been employed and the conclusions appear to be justified based on the data at hand. I have a few recommendations for consideration.
Introduction. Please provide a clear hypothesis to be tested in the study. Results. Can the authors analyze data according to sex for possible sex differences in outcomes? Results. Can the PM be measured in participants and also what is the identity of them? Results/Discussion. Can the authors provide actual number of CVEs as well as percentages (as already provided). Discussion. Can the authors suggest mechanisms for the PM and high temperature induced CVEs. A scheme depicting possible mechanisms would be helpful to the reader. General. Please check that manuscript format is in compliance with journal requirements.Author Response
Introduction. Please provide a clear hypothesis to be tested in the study.
We have reformulated the Introduction to render the study hypotheses explicit.
Results. Can the authors analyze data according to sex for possible sex differences in outcomes?
None of our analyses revealed a significant interaction between PM10 and sex or age. We state this in the Results section (line 300 et seq.).
Results. Can the PM be measured in participants and also what is the identity of them?
We were only concerned with levels of atmospheric PM and their health consequences in terms of CVEs, as we now state explicitly in the introductory hypotheses. We know the inhaled PM is efficiently absorbed by the body and can be observed in tissues (most easily lung tissue), but we were not concerned with quantifying PM in tissue samples. We do not feel that taking tissue or blood tissue samples would be ethically justified, particularly since methods of quantifying atmospheric PM are well developed.
Results/Discussion. Can the authors provide actual number of CVEs as well as percentages (as already provided).
We state in the first line of the Results “A total of 1349 A&E presentations for CVE were identified in the 2014-15 study period….”
Discussion. Can the authors suggest mechanisms for the PM and high temperature induced CVEs. A scheme depicting possible mechanisms would be helpful to the reader. General. “
We have rewritten the Discussion in the hope of clarifying the hypothesized mechanisms by which high temperature exacerbated the effect of high PM on presentations for CVEs. Briefly, these proposed mechanisms are: (a) variation in PM composition with temperature; (b) seasonal (hence temperature related) changes in behavior that influence the effect of high PM on presentations for CVEs; (c) general exacerbation of effects of toxins (ingested with PM) with high temperature.
Please check that manuscript format is in compliance with journal requirements.
Done.
Round 2
Reviewer 1 Report
None. Thank you for addressing the comments. The manuscript is much more complete.